

# Kinetic Grain Growth in Firn Induced by Meltwater Infiltration on the Greenland Ice Sheet

Kirsten L. Gehl[1], Joel T. Harper[1], Neil F. Humphrey[2]

[1]Department of Geosciences, University of Montana, Missoula, MT, 59801, USA
5   [2]Department of Geology & Geophysics, University of Wyoming, Laramie, WY, 82071, USA

*Correspondence to*: Kirsten L. Gehl (kirsten.gehl3798@gmail.com)

**Abstract.** The microstructure of polar firn governs its porosity, permeability, and compaction rate, and is thus critical to understanding surface elevation change, heat and gas exchange, and meltwater infiltration on ice sheets. Previous studies in high-elevation dry firn have identified two atmospheric drivers of kinetic grain growth, though both produce only millimetre-scale layers near the surface. Here, we demonstrate that meltwater infiltration in the percolation zone of the Greenland Ice Sheet (GrIS) produces centimetre- to decimetre-scale layers of kinetic grain forms, ranging from faceted crystals to depth hoar, persisting to depths of up to 16 m. We analysed subsurface temperature time series from a transect on the western GrIS to resolve thermal regimes associated with infiltration-driven kinetic grain growth. Two distinct mechanisms responsible for faceting were identified: one associated with the onset of the wet layer, the other with preferential meltwater flow events. For both mechanisms, elevated vapor fluxes were calculated and diminished grain sphericity was observed in SNOWPACK model simulations, implying each can facilitate kinetic grain growth. Wet layer onset was the dominant mechanism, producing pronounced reductions in sphericity and the most enduring faceted layers. Additionally, the rate of wetting front propagation influenced the longevity of faceted layers, with rapid infiltration preferentially producing lasting, lower-sphericity firn grains. As surface melt expands across the GrIS, constraining the influence of these faceted layers on meltwater storage, surface elevation change, and chemical transport will become increasingly important.

## 1 Introduction

The microstructure of snow and firn encompasses its optical, mechanical and physical characteristics, including the geometries of ice grains, bonds, and pore spaces (e.g., Colbeck, 1987), all of which are fundamental to understanding the evolutionary processes of the firn column. Key properties such as grain size and form influence the firn column's permeability and porosity (Adolph and Albert, 2014; Amory et al., 2024). Microstructural characteristics, therefore, govern the exchange of heat and mass (Adolph and Albert, 2014; Calonne et al., 2019) through the movement of air, water, and vapor (Davis et al., 1996), as well as the transport of chemical species (Johnsen et al., 2000). Furthermore, microstructure plays a critical role in the physics





controlling the rate of firn compaction (Anderson and Benson, 1963; Salamatin et al., 2009), an essential aspect of all studies concerning surface elevation change of ice sheets.

Persistent katabatic winds along the flanks of the Greenland Ice Sheet (GrIS) typically result in a deposited snowpack consisting of wind-packed, sub-millimetre, broken crystal fragments (e.g., Benson, 1960). Coupled mechanical and thermodynamic processes increase density with burial, eventually forming firn and, later, glacial ice (Cuffey and Paterson, 2010). In addition to grain deformation and mutual displacement, the size, shape, and bonding between grains evolves relatively rapidly. Molecular movement—via volume and surface diffusion—reduces curvature variation between grains and

facilitates grain rounding (Colbeck, 1980, 1982a; Yosida, 1955). Bonds enlarge at grain contacts, primarily driven by vapor transport via sublimation due to high vapor pressure over ice (Colbeck, 1980, 1983). Additionally, larger-radius grains grow at the expense of smaller ones, reducing total surface energy (Colbeck, 1980). After aging and burial to several metres' depth, firn grains become rounded with maximum diameters falling near 2 mm (e.g., Benson, 1960; Vandecrux et al., 2022).

Despite prevailing rounding processes in the firn column, observations of near-surface, centimetre-scale layers of depth hoar

on the Antarctic Ice Sheet (Albert et al., 2004; Gow, 1968; Grootes and Steig, 1992; Watanabe et al., 1997) and the Greenland Ice Sheet (Alley et al., 1990; Benson, 1960; Steffen et al., 1999) imply that kinetic grain growth can occur in some circumstances. Field and laboratory work has long established that the vapor flux favoured by strong temperature gradients facilitates kinetic grain growth and the formation of faceted grains (e.g., Akitaya, 1974; Colbeck, 1982b, 1983; Marbouty, 1980). The spectrum of resulting forms ranges from grains with squared sides to cup-shaped pyramids (Fierz et al., 2009; Fierz and Baunach, 2000) (the latter often called "depth hoar"). In the case of polar ice sheets, observed thin faceted layers are

attributed to strong temperature gradients that develop in low-density, near-surface snow in high-elevation, dry firn environments. These surface temperature gradients are driven by warm katabatic winds during winter or by solar insolation in summer (Albert et al., 2004; Alley et al., 1990; Steffen et al., 1999). Facets formed by these mechanisms are preserved up to ~2m depth before elimination by rounding processes and overburden-driven compaction processes.

Rare observations of highly developed faceted crystals occurring in centi- to decimetre-scale layers at depths of up to ~5 m have been reported in the GrIS percolation zone (McDowell et al., 2023; Nghiem et al., 2005). In addition, later in this paper we present detailed documentation of faceted crystals in firn cores from the GrIS percolation zone including individual layers of up to 1 m thick and identifiable as deep as 16 m below the surface. These findings are inconsistent with previously proposed mechanisms for GrIS kinetic growth, which are limited to thin, near-surface layers under cold and dry conditions (Albert et

al., 2004; Alley et al., 1990; Steffen et al., 1999). Instead, alternative mechanisms—potentially driven by the unique interplay





of wet and cold conditions in the percolation zone—may be responsible for the development and preservation of kinetic grain forms.

In the percolation zone, meltwater generated at the surface infiltrates the underlying firn, which remains significantly colder due to the preceding winter cold wave. Widespread meltwater forms a 0˚C wet layer that penetrates downward up to several meters depth and persists for weeks or more during the summer melt season (e.g., Saito et al., 2024). Additionally, preferential flow paths (pipes) locally route additional meltwater up to several meters below the wetting front (Colbeck, 1972; Humphrey et al., 2012; Marsh and Woo, 1984a, 1984b). Meltwater infiltration therefore advects heat into cold firn, potentially establishing localized temperature gradients sufficient to support kinetic grain growth. Kinetic grain growth has previously been linked to wet snow layers and melt-freeze crusts in laboratory studies (e.g., Hammonds et al., 2015; Jamieson and Fierz, 2004) and in seasonal snowpacks (e.g., Dick et al., 2023; Jamieson, 2006).

Here, we document extensive faceting within the firn column of the GrIS percolation zone. We investigate the mechanisms driving kinetic grain growth under the unique temperature gradients generated by meltwater infiltration into cold firn. Using nearly a decade of in situ temperature time-series data, we characterize temperature gradients and vapor fluxes associated with wetting fronts and preferential flow paths. To constrain kinetic growth under observed conditions, we apply the one-dimensional multi-layer firn model, SNOWPACK (Bartelt and Lehning, 2002; Lehning et al., 2002a, 2002b), to evaluate changes in grain sphericity. As surface melt intensifies across the GrIS, understanding microstructural evolution driven by meltwater infiltration processes is becoming increasingly critical for interpreting surface elevation change and assessing firn's capacity to store meltwater.

## 2 Methods

### 2.1 Study area and observation period

Field observations were collected along the Expéditions Glaciologiques Internationales au Groenland (EGIG) line (Finsterwalder, 1959) in the western GrIS percolation zone. The transect ranges from 1401—2102 m.a.s.l (Fig. 1) and data were collected from 2007—2009, 2017—2020, and 2022—2024. While melt across the transect decreases with increasing elevation, winter accumulation is relatively equal across the region, averaging about 0.5m w.e. yr$^{-1}$ (RACMO, Noël, 2019). In situ temperature measurements throughout the firn column reveal that the summer heat wave penetrates more deeply at lower transect elevations, driven by increased latent heat transfer and higher thermal conductivity resulting from greater firn density. These processes produce steeper temperature gradients in the deep firn, enhancing heat flux into the underlying ice (Saito et al., 2024).



## 2.2 Firn core measurements

A total of 38 firn cores with 79mm diameter were drilled to depths from 4-32m, with 95% of cores ≥10m and 50% of cores ≥25m depth. Cores were processed in the field to measure density, ice content, and microstructural characteristics including grain size, grain form, and specific surface area (SSA) on some occasions. SSA ($mm^{-1}$) measurements were collected using the infraSNOW device (Gergely et al., 2014) from which optical equivalent grain diameter (OED, mm) can be calculated (e.g., Gergely, 2011; Montpetit et al., 2011). Grain forms were categorized as faceted, depth hoar, rounded, or ice. Core sections

were weighed with an electronic balance with 1 gram resolution and lengths were measured with 1cm resolution, providing mean density of sections.

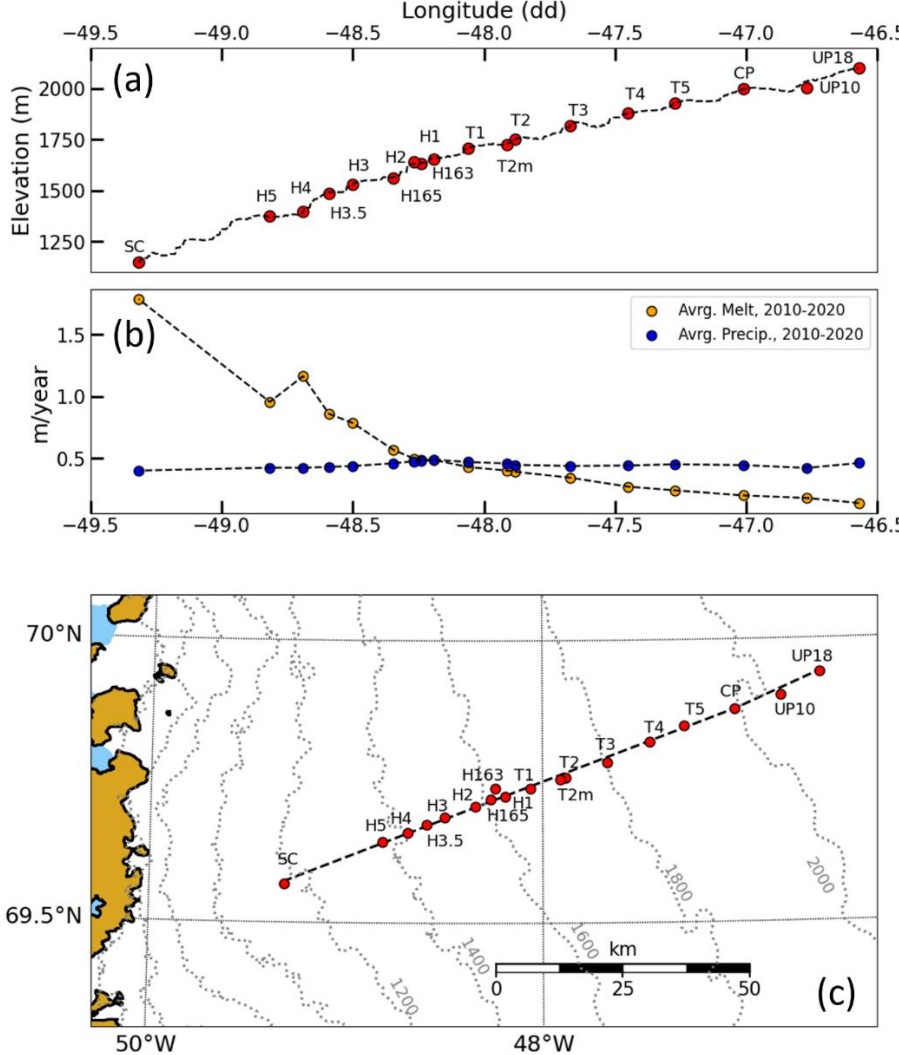

**Figure 1. Study area. A) Elevation profile, B) annual melt and precipitation amounts, and C) plan-view map of sites in the study area. SC (Swiss Camp) is not a part of this study but is the lowest elevation site on the EGIG. Precipitation and melt data were**
**averaged from year-resolution RACMO datasets from 2010—2020.**



## 2.3 Kinetic grain growth

Strong temperature gradients produce large vapor pressure gradients which drive mass transfer between snow grains and facilitate kinetic grain growth (Colbeck, 1982b, 1982a; Perla and Ommanney, 1985). Higher snowpack temperatures allow for greater vapor pressures and potential crystal growth, with vapor concentrations at -15℃ almost 1000x larger than at -65℃

(Kamata et al., 1999). However, measuring vapor pressure is challenging and prone to high error, thus it is widely accepted that working with temperature is best practice (Colbeck, 1987). We therefore adopt the approach of numerous prior studies (e.g., Colbeck, 1987; LaChapelle and Armstrong, 1977; Marbouty, 1980) that associate grain faceting with a threshold temperature gradient of $|\pm10|$ °C m$^{-1}$. Here, we use the term Critical Temperature Gradient (CTG) to refer to any temperature gradient above the threshold that is therefore indicative of facet-forming conditions.

### 2.3.1 Temperature gradients

Temperature strings were installed in 24 core holes at 17 sites (Fig. 1) over the observational record. Holes were backfilled with fine-grained cold snow following installation. Strings installed in 2019 and 2022-2024 were fitted with digital temperature chips spaced from 0.125—1m apart, depending on year, with accuracy of 0.1 °C and resolution of 0.0078 °C. Thermistor strings were installed from 2007-2009 with sealed 50k ohm thermistors at nominal 0.5 m spacing. Because thermistors are

more sensitive to calibration and drift, accuracy is estimated at <0.5 °C with measurement precision of 0.02 °C. Data were recorded every 20-30 minutes (depending on location and year) by data loggers installed on poles at the surface. Given staggered installation and data downloading, time series cover varying intervals.

Central-differenced temperature gradients ($\partial T/\partial z$) were calculated at single timesteps from temperature time series. Gradients exceeding the $|\pm10|$ °C m$^{-1}$ threshold were identified and categorized as a CTG. The CTGs associated with either a) wetting

fronts or b) preferential flow mechanisms of infiltration were classified by manual inspection of temperature time series, based on positioning and duration. For example, wetting front CTGs were immediately beneath the 0°C, surface-propagating isotherm and persisted for weeks, whereas the CTGs associated with piping events were far below the wetting front and lasted for hours to days. Site conditions and interannual variability of meltwater percolation also contributed to variability in the depths and durations of CTGs. In some cases, larger-magnitude gradients $\geq|-20|$ °C m$^{-1}$ were subset for comparisons.

### 2.3.2 Vapor fluxes

Provided our study sites are colder than typical seasonal snowpacks where most temperature gradient metamorphism studies are performed (e.g., the mean annual air temperature at CP-1998m is about -16.5℃), we expect relatively reduced rates of kinetic grain growth in GrIS percolation zone firn. We therefore calculate vapor fluxes from temperature to assure that temperature gradients are sufficient to drive vapor flux amidst cool ambient temperatures. Using the Clausius-Clapeyron

relation and the non-linear dependence of equilibrium water vapor density on temperature, vapor flux can be predicted from temperature (Colbeck, 1983; Sturm and Benson, 1997). We calculate equilibrium vapor density using Sturm and Benson



(1997)'s 1D vapor diffusion model for a subarctic snowpack, which assumes a constant thermal conductivity with snowpack height, heat/vapor flow only in the upwards direction, and that vapor density is in thermal equilibrium with the surrounding snow, a common assumption (Colbeck, 1982b). Approximate equilibrium vapor density can then be expressed as:

$$\rho_v = \rho_{v0} e^{[L(T-T_0)/(RTT_0)]},$$
(1)

where $\rho_{v0}$ is the equilibrium vapor density at the melting point (4.847e$^{-3}$ kg m$^{-3}$), L is the latent heat of sublimation (2.83e$^{-5}$ J kg$^{-1}$), $T_0$ = 273.15K, T is the measured temperature (Kelvin), and R is a gas constant (461.9 J kg$^{-1}$ K$^{-1}$). From Eq. (1), we can then solve for vapor flux using Fick's First Law (Colbeck, 1983; Pinzer et al., 2012):

$$J_v = -D_{eff}\left(\frac{\partial \rho_v}{\partial z}\right),$$
(2)

where $J_v$ is the vapor flux (kg m$^{-2}$ s$^{-1}$), $D_{eff}$ is the effective vapor diffusivity in the air component of snow (8e$^{-5}$ m$^2$ s$^{-2}$) (Pinzer et al., 2012), and $\frac{\partial \rho_v}{\partial z}$ is the equilibrium vapor density gradient between two neighboring temperature sensors within a temperature profile. Because temperatures in our study area are like those of a subarctic snowpack, we compared modelled vapor fluxes to those reported by Sturm and Benson (1997).

**2.4 SNOWPACK simulations**

To constrain sphericity changes in grain forms due to infiltration driven metamorphism, we utilized the 1D multi-layer snow physics model, SNOWPACK (Bartelt and Lehning, 2002; Lehning et al., 2002a; Lehning et al., 2002b). SNOWPACK performs reliably under conditions where meltwater is present (Wever et al., 2015, 2016), making it suitable for simulating the percolation zone. SNOWPACK simulates grain sphericity evolution over time – the most quantitative indicator of kinetic grain growth, as facets exhibit low sphericity – enabling us to test the effectiveness of various meltwater-induced temperature

regimes in reducing sphericity. A site-specific simulation was run based on weather and firn stratigraphy data to evaluate temperature gradient metamorphism under observed conditions at the CP-1998m site. Additionally, idealized simulations were run to test two observed temperature gradient scenarios (described in the Results section) independent of site-specific ice layer stratigraphy or weather events.

**2.4.1 Site-specific simulation**

We simulated a summer melt season at site CP-1998m to assess sphericity changes in a complex firn structure subjected to highly variable and transient atmospheric forcing. The model was run from 1-March, 2019—31-December, 2019 and forced with data from the ECMWF Reanalysis v5 (ERA5) climate model (Hersbach et al., 2020), as in-situ GC-NET PROMICE (Steffen et al., 2022) weather data at the site in 2019 lacked incoming LWR values. ERA5 forcings were provided at hourly timesteps.



We initialized the model with spring conditions using data from a firn density profile measured in a core drilled in May 2023
and a temperature profile measured on 1-March, 2024. Microstructural properties ascribed to the firn profile are provided in
Table S1. SNOWPACK was configured in its "Polar" variant (Steger et al., 2017), utilizing Richard's equation for water
transport (Lehning et al., 2002a; Wever et al., 2015), and with mass transport by vapor flow (Lehning et al., 2002a).
Precipitation was set to reaccumulate at each timestep and internal timesteps were set to 20 minutes with outputs every hour.

**2.4.2 Synthetic simulations**

To investigate whether meltwater can systematically induce kinetic grain growth and faceting, we conducted two synthetic
firn column simulations under idealized initial conditions and meteorological forcings. Both simulations were initialized using
a 50-year spin-up with sinusoidal, annual-wavelength meteorological forcings that excluded melt, but with a mean annual
temperature representative of conditions at ~2000 m elevation along our study transect. The resulting firn column following
spin-up on April 1st, for example, exhibited a near-vertical deep temperature profile of −16.5 °C and a smooth density profile.
SNOWPACK was configured identically to the site-specific simulation during the spin-up and additional parameterizations
are provided in the Supplement (Table S2, Fig. S1).

The first of the two transient synthetic simulations, referred to as the Wetting Front simulation, modelled the downward
propagation of the 0 °C isotherm over the summer melt season. Air temperature and shortwave/longwave radiation forcings
were applied at hourly timesteps, using average values measured from 1995–2020 by the GC-NET PROMICE station at CP-
1998m (Fig. S1) (Steffen et al., 2022). The simulation was run from 1-April—31-December. The second simulation, termed
the Preferential Flow (Piping) simulation, was designed to reproduce a thermal signature characteristic of a piping event, where
isolated deep warming results from liquid water traveling through preferential flow paths and refreezing above an impermeable
layer. Because preferential flow physics are not well represented in SNOWPACK, a simplified workaround was employed by
inserting a ~10 cm-thick layer composed of 13% liquid water, 85% ice, and 2% air, between 2.0–2.1 m depth and directly
above a 1 cm-thick, 100% ice layer. As melt was not produced during the spin-up, grain radii were universally smaller than
those typically observed in the percolation zone. To address this, we uniformly increased grain radii by a factor of 1.5, resulting
in average sizes slightly smaller than those reported by McDowell et al. (2023). Initial sphericity values were also modified
and set to 0.5 throughout the firn column. This simulation was run for 15 days (1–15 July) using the same meteorological
forcings as the Wetting Front simulation (Fig. S1), but with 10-minute internal timesteps to better resolve rapid thermal
responses.



# 3 Results

## 3.1 Faceted Grains

We observed faceted grains ranging from endmember, hexagonally-shaped depth hoar to, more commonly, square-shaped
grains or grains with angular sides (Fig. 2). When viewed in core sections, faceted layers were distinct due to their rough,
jagged outer surface (Fig. 2). In some circumstances, faceted layers were friable and disintegrated in the core drill.

Faceted grain diameters ranged from 1—5.5mm, with a mean diameter of 2.6mm (n=231). At 11.6m depth at site UP10-
2005m, the optical equivalent diameter (OED) of faceted firn grains ranged from 1.15—2.4mm, with a mean OED of 1.33mm.
Our faceted grain sizes and OED values were larger than the grain diameters of ~0.9—1.2mm observed at high-elevation
percolation zone sites by McDowell et al. (2023), or the 0.8—1.7mm OED values in a dry firn core drilled at Summit Station
(Linow et al., 2012). Facets were identified at 9 sites along the transect and within 19 individual cores (Fig. 3). Generally,
faceted layers were less noticeable and more challenging to identify with increasing depth. Faceted layers were observed from
0 to 16m depth with thicknesses of up to 1m (Fig. 3). Most faceted layers were located within 5m of the surface, but notable
layers at 6 sites were identified >10m below surface (Fig. 3).

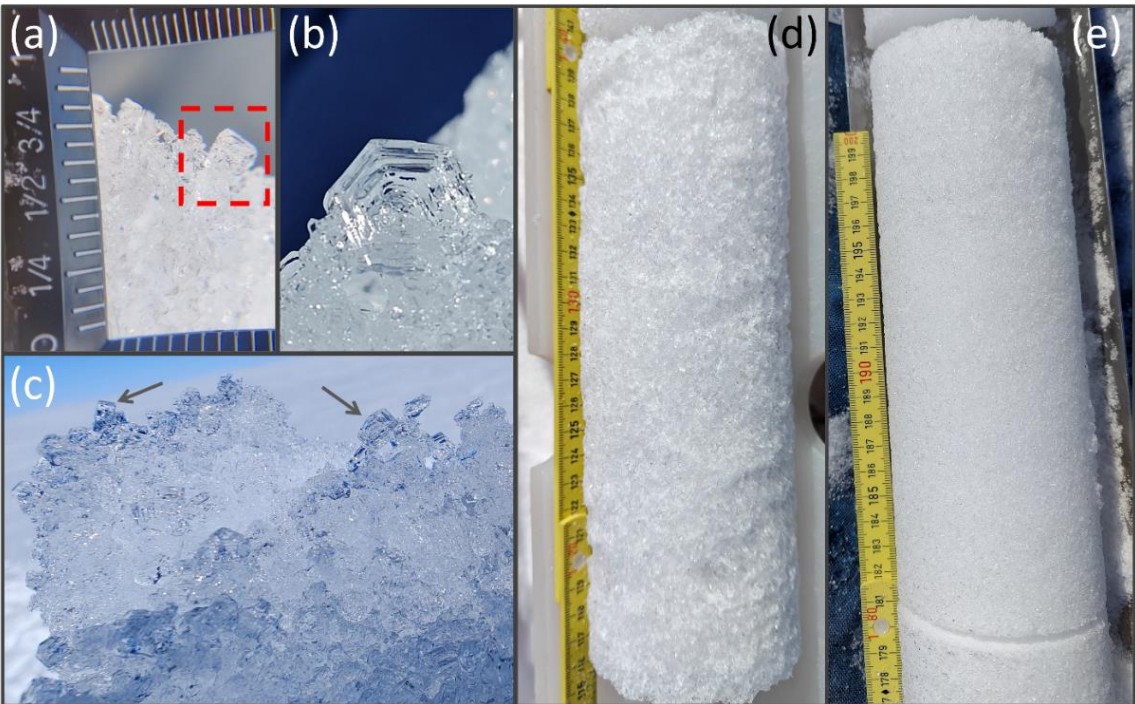

195

**Figure 2. Photographs of faceted grains. A) Square, faceted firn grain observed at site T3-1818m at 6.9m depth, spring 2022. The
straight, upward-facing edge of the grain within the red dashed box measures ~3-4mm in length (see scale). B) Depth hoar grain
observed at site CP-1998m at 1.5m depth, spring 2024. The grain was located beneath the previous year's summer melt surface, with
an estimated grain size of ~4mm. This grain shape represents an end-member kinetic growth form. C) Example of friable, faceted
200 grain clusters pulled from a core section. Arrows point to two square, faceted grains. D) Coarse, faceted firn grains within a rough,
bumpy core section (site CP-1998m, spring 2023, 11m depth), compared to E), fine, ~1mm diameter rounded firn grains in core
section (site CP-1998m, spring 2022, 10m depth).**



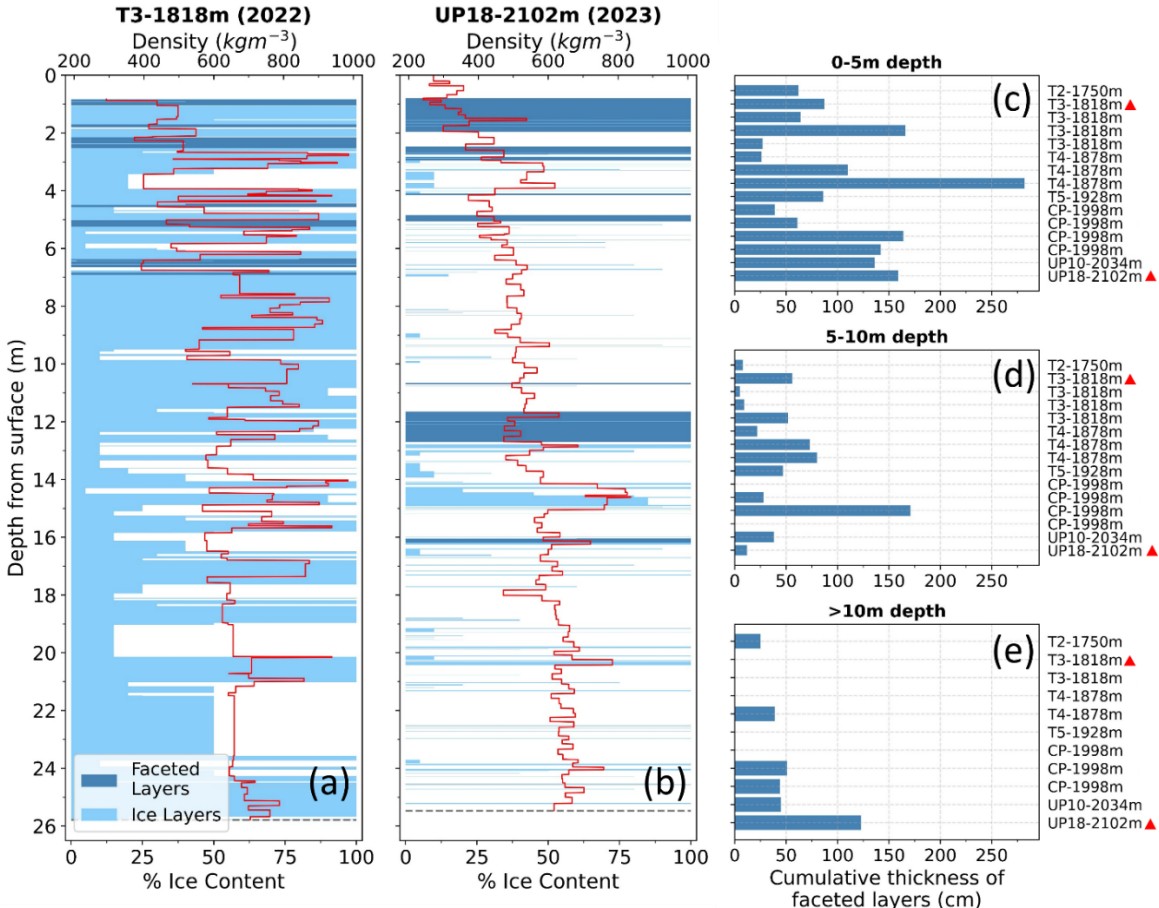

**Figure 3. Firn core profiles and depth distribution of observed faceted layers. A) & B) Ice content, faceted layers, and density stratigraphy from two cores: A) T3-1818m, drilled 2022 and B) UP18-2102m, drilled 2023, where light blue horizontal bars indicate approximate % ice content and dark blue bars indicate faceted layers. Density at a given point in the firn core is denoted by red lines. C, D, E) Cumulative thickness of observed faceted layers in individual cores from C) 0-5m depth, D) 5-10m depth and E) >10m depth. Cores are labelled by 'Site I.D.-Elevation' and are in order of elevation; repeated labels imply multiple cores were drilled at the site in different years. Cores with red triangles next to their names are displayed in A) & B).**

## 3.2 Temperature gradients

The temperature gradients measured in our network of time series exceeded the critical threshold of $|\pm10|$ °C m$^{-1}$ in most years with measurements and at numerous depths and locations. CTGs were measured in 23 individual core holes and at 14 locations along the transect, and from 0—12.5m depth below surface. CTG magnitudes ranged from -159 to -10°C m$^{-1}$ and from 10 to 132°C m$^{-1}$, with the strongest CTGs occurring closer to surface (Fig. 4). The depths at which CTGs occurred varied with season, with CTGs seen anywhere from 0-12.5m depth June—September and from 0-2m depth September—May. While seasonal temperature swings produced temperature gradients in the firn column (Fig. 5a), gradient magnitudes were typically sub-threshold, except near the surface in May at one site (Fig. 5b). The shallow CTG signal was due to diurnal air temperature fluctuations, which were also observed for shorter periods at other sites and times of year.





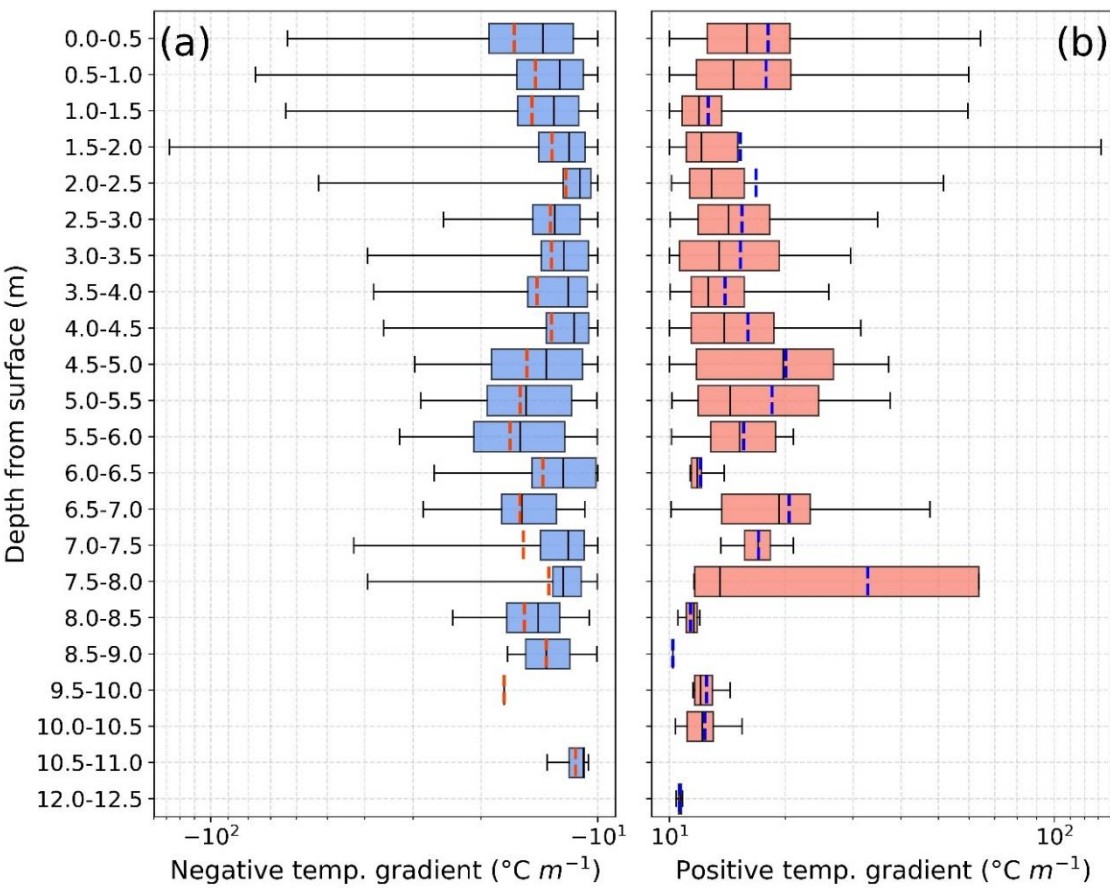

**Figure 4. Box and whisker plots of both A) negative and B) positive CTG magnitudes and their frequencies over the observational records. The x-axes use a logarithmic scale for values < |±100| and a linear scale for values > |±100|. Whiskers cover the entire range of each dataset. Red or blue vertical bars within boxes denote means; black vertical lines denote medians. Sample sizes for each box are not equal and decrease with depth.**

The two mechanisms of meltwater infiltration, (1) the onset of a surface wet layer, and (2) preferential flow/piping, generated temperature gradients conducive to kinetic grain growth (CTGs). Additionally, a third recurring pattern of CTGs was linked to near-surface diurnal temperature fluctuations; however, due to their short duration and the likelihood of subsequent wet metamorphism, we do not further consider diurnal processes. CTGs associated with wet layers and piping were identified in 17 of 24 sites (Fig. 6, Table S3), all of which experienced summer melt, indicating that meltwater infiltration alone was insufficient to generate gradients above the threshold. We structure further analysis of kinetic grain growth around the two primary meltwater-driven mechanisms.



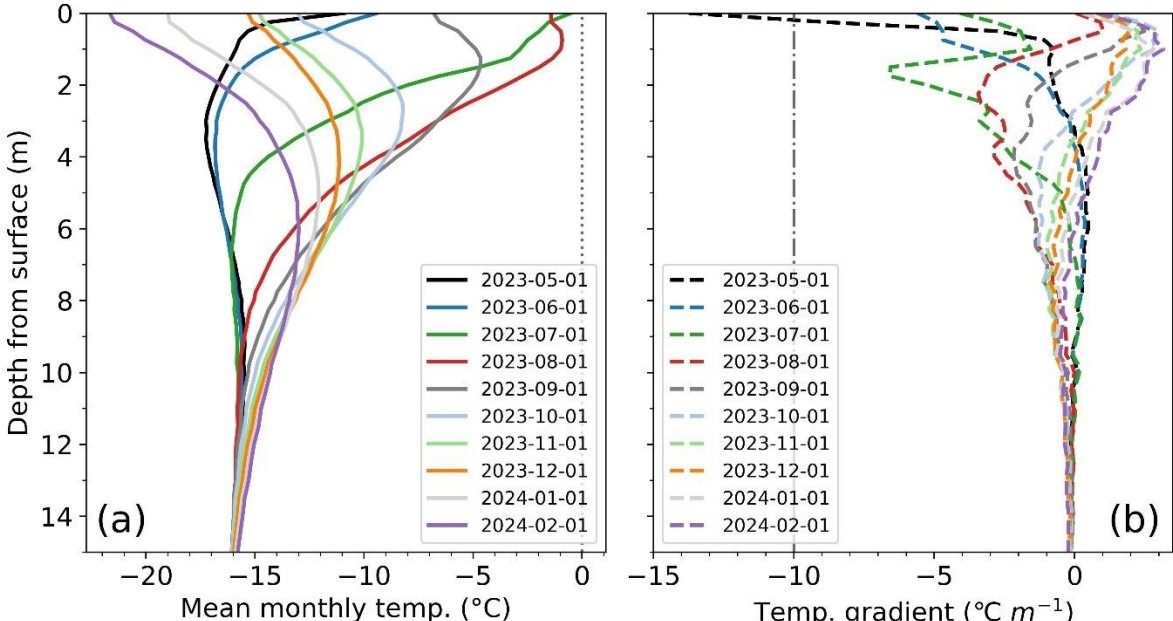

**Figure 5. Example of a seasonal firn temperature cycle at site CP-1998m (May 2023-February 2024). A) Mean monthly temperature profiles and B) temperature gradients calculated from monthly means. Profiles in A) and B) were assigned first-of-month dates. The dotted line in A) denotes 0°C and the dashed-dot line in B) the negative CTG threshold. Monthly average temperature gradients in B) only surpass the CTG threshold in May (black line) near-surface.**

### 3.3 Wet layer onset

Meltwater infiltration via surface wetting fronts reached depths ranging from approximately 0 to 4 meters. Wetting fronts tended to advance as stepped, rapid pulses over the course of several days, penetrating from decimetres to several metres. Only the highest elevation sites during the coldest years lacked significant wetting fronts. In most cases, a single distinct wet layer formed per melt season; however, in rare instances, a second wetting front developed following the refreezing of the first, triggered by a late-season warm spell or rain event (e.g., Harper et al., 2023).

The large temperature contrast between the 0 °C isotherm at the wetting front and the underlying cold firn produced strong, negative-signed temperature gradients (Fig. 7). Critically, CTGs associated with wetting fronts tended to coincide with the onset of infiltration rather than persisting throughout the entire life of the wetting front. The depth of wetting-front CTG formation varied by site and year, ranging from 0.13 to 5 m (Fig. 6). Cumulative CTG durations spanned from 57 to 703 hours, with a mean of 264 hours (Table S3). CTG duration also varied with elevation, with notably shorter durations occurring at lower-elevation sites (Fig. 6). The longest durations—recorded at sites T3-1818m, CP-1998m, and UP18-2102m (Fig. 6)— were comparable to the durations of above-threshold temperature gradients known to produce depth hoar in seasonal snowpacks (Giddings and LaChapelle, 1962). In addition, vapor fluxes produced by wet layer onset were substantial, ranging



from ~0.5 to 4.5e$^{-7}$ kg m$^{-2}$ s$^{-1}$, mirroring temperature gradient distribution (Fig. 7) and comparable in magnitude to those observed in a cold, subarctic snowpack with a large temperature gradient that produced depth hoar (Sturm and Benson, 1997).

Modelling results demonstrate that rapid wetting front descent is critical for generating enduring layers of kinetic grain growth. In the synthetic simulation, the wetting front descended slowly over ~15 days—compared to an observed average of 3.4 days at site CP-1998m—and produced CTGs lasting 1101 hours, falling below −20 °C m$^{-1}$ for 25% of that time (Fig. 8). Strong
sphericity reductions occurred before and during descent, rather than beneath the wetting front's maximum depth, but were largely reversed by wet snow metamorphism as the slowly descending front overtook earlier-formed facets. Furthermore, sphericity reduction that did occur below the maximum wetting front extent subsequently re-rounded. These results indicate that rapid wetting front descent is necessary to prevent the overprinting of kinetic grain forms.

The site-specific simulation produced two wet layers that descended more rapidly, over ~2–4.5 days, generating CTGs lasting
894 hours that were lower than −20 °C m$^{-1}$ for 26% of the time (Fig. 7). In this case, sphericity beneath the maximum extent of the wetting front declined from 0.5 to 0.35 and did not re-round, resulting in a layer of kinetic grains that persisted for the remainder of the simulation. Although both simulations produced similarly strong CTGs, those in the site-specific case were more spatially confined and temporally persistent, leading to (1) more localized kinetic grain growth (compared to broader sphericity reductions in the synthetic simulation) and (2) layer preservation in the firn column.






**Figure 6. Duration, timing, and depth characteristics of meltwater-induced faceting mechanisms, wet layer onset (left column) and preferential flow (right column), at each site (where observed). A & B) Cumulative time that critical temperature gradients (CTGs) were observed at each site. C & D) Range of depths where CTGs associated with mechanisms were observed. E) Time periods where CTGs associated with wet layers were observed. F) Dates where CTGs associated with preferential flow events were observed.**



**Figure 7. Observations through time from site CP-1998m (2023) showcasing wet layer onset (left column) and preferential flow (right column) faceting mechanisms. The black box in C) highlights data displayed in B, D, & F. Grey lines denote the general extent of the contiguous wet layer at a given time. A, B) Contour plots of ambient firn temperatures. C, D) Contour plots of temperatures gradients. Blue and red regions indicate CTGs. E, F) Contour plots of calculated vapor fluxes.**






**Figure 8. Model outputs of temperature gradient (left column) and sphericity time series (right column): A & B) synthetic wet layer simulation, C & D) site-specific simulation, and E & F) synthetic preferential flow simulation. Zero-degree regions in A—D) are denoted by black hatching; approximate wetting front position is indicated by black lines with white outlines. The depth ranges where the liquid water layer was input in E & F) are indicated by black hatching.**



## 3.4 Preferential flow/piping

Our temperature time series showed numerous, short-duration episodes of near-0°C temperatures that were localized beneath and disconnected from the surface wet layer (Fig. 7), which we attribute to latent heat release due to preferential flow infiltration events. These events spanned narrow depth intervals between 0.75 and 12.5 m below surface (Fig. 6), and generated CTGs of ranging from 2 to 190 hours. The refreezing at depth occurred within a temperature field that warmed toward the surface, resulting in both positive (above) and negative (below) CTGs around the 0 °C layer. Vapor fluxes during the events were also very high (≥$4.5 \times 10^{-7}$ kg m$^{-2}$ s$^{-1}$), but relatively short-lived (Fig. 7).

Our synthetic piping simulation revealed a limited degree of kinetic grain growth due to a typical piping event (Fig. 8). A 10 cm-thick liquid water layer introduced at 2 m depth generated CTGs spanning ~1.0 m of the firn column, from ~1.6 to 2.6 m. Positive CTGs formed above the refreezing layer, lasting ~72 hours and spanning ~0.25 m, while negative CTGs developed below, lasting ~120 hours and spanning ~0.5 m. These durations are consistent with observed CTGs associated with piping events (Fig. 6). Similar sphericity reductions, modestly decreasing from 0.5 to ~0.45, took place above and below the refreezing layer.

## 4 Discussion

Our findings indicate that kinetic grain growth in the GrIS percolation zone is primarily driven by the onset and initial descent of surface wetting fronts, which juxtapose the 0°C isotherm with underlying cold firn (Figs. 6, 7). Rapid infiltration produces strong thermal contrasts, whereas slower wetting front advancement allows conductive heat transfer to warm the firn below, thereby reducing the strength and duration of temperature gradients (Fig. 8). Fast penetration of wetting fronts is common in our dataset, facilitated by abrupt onset of warm air temperatures as well as preconditioning of firn temperature by preferential piping events, which locally warms the firn causing faster vertical propagation of the wetting front.

While our study identified specific scenarios for meltwater-induced faceting, additional processes may also contribute to kinetic grain growth and the preservation of faceted grains in the firn column. For instance, a sudden and prolonged cold snap in late summer could expose the warm, rounded grains in the surface wet layer to a cold atmosphere, causing partial or complete faceting. However, preservation is critical: faceted grains formed near the surface in one year may be overprinted by melt and rounding in the following melt season. Interannual variability is therefore a key factor—preservation of faceted layers is most likely when a year of strong faceting is followed by minimal melt the next year.

Our observations indicate that conditions conducive to kinetic grain growth are induced by meltwater infiltration, which might suggest that increases in the extent, duration, and intensity of surface melt on ice sheets enhance faceting in firn layers. However, our findings show that infiltration-induced faceting is not a simple or direct function of melt amount (i.e., Figs. 3



and 6). Antecedent firn microstructure plays a critical role in determining where and when meltwater is retained (e.g., Marsh
Woo, 1984a; Humphrey et al., 2012), and influences both the rate of wetting front propagation and the partitioning of water
between matrix and preferential flow pathways (Pfeffer and Humphrey, 1996; Moure et al., 2023). The thermal state of the
firn is also a key factor: we observed less faceting at lower elevations, despite higher melt totals, because the temperature
contrast between the 0°C isotherm and relatively warmer firn was less pronounced. Taken together, these findings suggest that
as surface melt intensifies and migrates to higher elevations, the firn conditions most favorable for kinetic grain growth may
similarly shift upward.

Widely used firn densification models (e.g., Herron and Langway, 1980) are based on empirical relationships derived from
dry firn conditions and do not explicitly incorporate the mechanical behavior or densification mechanisms of faceted grains
and depth hoar, which differ substantially from those of rounded grains (Fourteau et al., 2024; Hagenmuller et al., 2015;
Hirashima et al., 2011). As melt-altered firn microstructures become more prevalent, the validity of applying these models to
interpret surface elevation changes across increasing portions of the GrIS is increasingly uncertain. Furthermore, as interpreting
melt-affected ice and firn cores becomes more common (Moser et al., 2024), understanding the effects of kinetic grain growth
on isotopic fractionation is growing in importance. The presence of faceted layers may also alter meltwater infiltration
processes, as these grains affect firn capillary properties and porosity (e.g., Amory et al., 2024; Colbeck, 1974; Zermatten et
al., 2014), which may regulate subsequent infiltration and, in turn, ensuing faceting.

## 5 Conclusions

This study presents extensive observations of faceted crystals within the firn column of the GrIS percolation zone, ranging
from squared forms to depth hoar, found at depths up to 16 meters, and in layers with thickness up to 1 meter. These findings
are incompatible with previous explanations of kinetic grain growth in firn, which were limited to thin, near-surface layers in
cold and dry environments. We explored the mechanisms responsible for forming these faceted grains using a variety of
approaches, including analysing nearly a decade of in-situ temperature time series data, calculating vapor fluxes, and
conducting model simulations of grain growth.

Our analysis identified two primary mechanisms for meltwater-induced kinetic grain growth: surface wet layer penetration
and preferential flow events. Both produced critical temperature gradients and elevated vapor fluxes sufficient to drive faceting,
with wet layer onset affecting a larger portion of the firn column for longer durations, resulting in more kinetic grain growth.
The rate of wetting front propagation strongly influenced the establishment of faceted layers in the firn column, as rapid
infiltration promoted the formation and, critically, the preservation of low-sphericity grains. However, faceting was not a direct
function of melt amount; rather, it depended on the interplay between firn microstructure, thermal state, and infiltration
processes. These findings have implications for modelling surface elevation change across much of Greenland and parts of

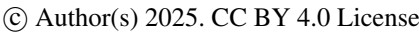



Antarctica, as empirically based firn densification models do not account for the distinct mechanical properties of firn grains
impacted by meltwater processes.

**Data Availability**

All firn core and temperature data presented in this manuscript are archived with full metadata description and are available for public download from Harper et al. (2012) or the N.S.F. Arctic Data Centre (Harper and Humphrey, 2023, 2024a, 2024b, 2024c, 2024d).

**Author Contribution**

JH and NH acquired funding for this research. All authors aided in data acquisition and processing, as well as in grain form assessment. KG and JH conceptualized temperature and temperature gradient analyses, while KG carried them out. KG conceptualized and designed code for vapor flux calculations. KG and JH conceptualized SNOWPACK model simulations; KG designed model code and operated model simulations. All interpretations (raw data, SNOWPACK results) were made by
KG and JH. Data visualization and initial draft preparation was completed by KG. Drafts (prior to article registration) were edited and reviewed by all authors.

**Acknowledgements**

The authors extend their gratitude to the many individuals who have contributed to the field work involved in data collection over the years.

**Financial Support**

This research was funded by the U.S. National Science Foundation, Award #2113391.

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
