# Peer review of "Kinetic Grain Growth in Firn Induced by Meltwater Infiltration on the Greenland Ice Sheet"

_EGUsphere, 2025_

## Author Response (AR1)

We thank the two reviewers for detailed and thoughtful comments on our original submission. We believe that addressing their reviews has substantially improved the manuscript. Below, we show each comment and a summary of how we have addressed the comment in our revision to the manuscript ("> *in blue*"). We have also provided an updated version of the manuscript and Supplement that uses Word's 'track changes' to show our edits. During the response process, we noticed mislabeled table numbers in the Supplement (independent of reviewer comments) and have made the appropriate changes to fix this.

**Reviewer**: Peter Kuipers Munneke

**Comments to the Author**
General Remarks

In this manuscript, an explanation for the occurrence of faceted grains in Greenland firn up to a depth of 16 metres is sought. Traditional mechanisms of faceted grain formation focus on processes close to the surface and in cold locations, but none are able to explain the formation and occurrence of faceted crystals deeper down in the firn, and in locations where frequent surface melt occurs. As far as I can judge, this is the first attempt to explain faceted grain formation in the Greenland percolation zone. It's an interesting and original subject, fit for The Cryosphere and its readership.

The central hypothesis is that meltwater infiltration (either as a wetting front or by means of preferential flow in pipes) creates very strong temperature gradients within the firn, between the temperate wet snow and the cold firn below the wetting front or surrounding pipes of preferential flow. Such temperature gradients act as hotspots for kinetic growth of faceted crystals.

This is an interesting hypothesis, made plausible by SNOWPACK modelling and by an inventory of in-situ observations of faceted crystals spanning a large range of elevations and time.

I do have some concerns about (1) the presentation and analysis of the observational data; (2) the modelling setup; and (3) the effectiveness of the figures. It is unlikely that these concerns will wipe out the central hypothesis of the paper. But if care is taken to clarify and work out a few points, it will make for a better and even more convincing paper.

> The comments discussed above in 1, 2, and 3 are answered in detail in subsequent responses.

**(1)** analysis of observational data

A critical temperature gradient (CTG) is introduced, defined as a temperature gradient whose absolute value equals 10 deg C m^-1. My feeling is that the statistics about the exceedance of this CTG in the observational record (presented for example in figures 4 and 6, and section 3.2) depend quite a bit on the vertical resolution of the temperature observations. Firn temperature is recorded with a vertical resolution between 0.125 and 1 m, limiting the ability to observe the highest temperature gradients. **How does vertical resolution affect TG observations? How**

**are the statistics in figure 6 affected by it?** If you downsample a record at 0.125m resolution to 1m,

You are correct to point out that vertical resolution is an important factor in our representation of temperature gradients. However, because heat transfer is driven by the diffusion process, smaller gradients at centimeter scales between our sensors will diffuse away on a relatively short time scales, limiting their impact in forming facets. Regarding dT/dz calculation from variable sensor spacings: we emphasize that dT/dz values are z normalized, meaning a CTG calculated and reported in °C/m accounts for sensor spacings <1m. Nevertheless, we cannot rule out finer scale facet-forming conditions for shorter time intervals – this makes our results, which are upscaled to a max of 1m resolution, a meaningful minimum assessment. Our purpose is to demonstrate with observational data that facet forming conditions indeed exist. As this is an important point to make clear to the reader, we have added to text to explain the implications and limits of the vertical resolution between sensors.

The statistics in figure 6 are a bit unclear to me. **Is the cumulative CTG duration (panels 6a and 6b) specified per year? Or in any other way normalized?** If the duration depends on the total length of the data set and/or the completeness of the data, then how to compare apples to apples?

> Our aim in showing the cumulative values of CTGs in 6a and 6b is to demonstrate the relative abundance of conditions favorable to kinetic grain growth measured with an observational network (c.f., inferred from model output). However, since the gradients are driven by infiltration processes, which are known to have high time/space heterogeneity, the values we measured are not necessarily representative of all places and all times. Yet, we feel it is important to present these numbers to demonstrate that real measurements show kinetic grain growth conditions exist in the firn and are present for extended durations sufficient for faceting.

Each value is labeled by site locations/years (rather than elevation or longitude) to signify the uniqueness of each observation, and the bottom panel shows the duration and timing of each time series. Thus, the purpose is not to compare one to another, but to demonstrate the abundance of facet forming conditions revealed by numerous samplings. Also, normalizing may imply that piping has a systematic relationship with wet layer development. This is the focus of other research and beyond the scope of this paper, but sadly neither we nor other workers have yet to uncover a functional relationship for partitioning between wet layers and piping. Having said all this, the comment reveals that we needed to do a better job describing the figure and explaining its purpose and limitations. We have added to the text and figure caption to achieve this.

In general, I feel that the occurrence of faceted grains is nicely presented in figures 2 and 3, and the statistics of large TGs in the subsequent figures. But is there a way to connect these two types of observations more precisely, if only for a case study or two? I.e., tying together observations of faceted grains in a particular firn core to the TGs of the preceding year at that

location? Such an illustration would strengthen the observational link between TG and faceted grains.

> This would clearly be a 'gold standard' test, but sadly our dataset does not permit such an analysis. The discovery of facets was serendipitous, and the field data collection program was not designed around documenting their growth. The field program required first drilling a core hole and logging the physical aspects of the core. The core hole was then fitted with temperature strings. This process was repeated at new sites each year such that the measurements of facets always come before the measurements of temperature. Further, our analyses show that the design of any future field campaign aiming to measure 'before' and 'after' conditions would need to work through several challenges including high interannual variability, variable overprinting of facets, difficulty tracking time horizons in a firn column with heavy infiltration and refreezing, and high spatial variability related to infiltration processes.

**(2)** the SNOWPACK modelling setup

When introducing SNOWPACK in section 2.4, it remains unclear what physics are implemented in the model that ensure that faceted grain growth is simulated reliably. **How is sphericity evolution modelled?**

> The SNOWPACK microstructure routine is well documented in Lehning et al. (2002). To begin, sphericity changes in SNOWPACK occur differently depending on whether snow is: 1) fresh new snow that typically undergoes rapid reduction in dendricity post deposition, 2) old, dry snow, in which sphericity change is primarily dictated by local temperature/vapor pressure gradients, or 3), wet snow. For old, dry snow (the type that we are concerned with), sphericity change through time occurs in two regimes dictated by $|\frac{\partial T}{\partial z}|$ and, secondarily, ambient snowpack temperature (T, below). The two regimes of sphericity change in old, dry snow are delineated by the following expression (Lehning et al., 2002):

$$sp\dot{}(t) = \begin{cases} 5 \times 10^8 e^{\frac{-6000}{T}} \left(5 - \left|\frac{\partial T}{\partial z}\right|\right), & |\frac{\partial T}{\partial z}| \leq 5\frac{K}{m} \\ -1 \times 10^8 e^{\frac{-6000}{T}} \left|\frac{\partial T}{\partial z}\right|^{0.4}, & |\frac{\partial T}{\partial z}| > 5\frac{K}{m} \end{cases}$$

Greater declines in sphericity (more kinetic growth) will occur at an ambient temperature of -1°C than at -20°C under a CTG of equivalent magnitude, according to the formulation. However, the paper directly states that the transition between the two $|\frac{\partial T}{\partial z}|$-dependent regimes is an arbitrary decision "not supported by theoretical considerations or observational data" (Lehning et al., 2002). Nevertheless, a very wide body of literature has established that the habit and degree of kinetic grain growth is driven by vapor flux set by the ambient temperature and $|\frac{\partial T}{\partial z}|$ (e.g., Akitaya, 1974), and that temperature gradients offer the best representation of metamorphic potential (e.g., Colbeck, 1987).

While an in-depth treatment of the processes and modeling parameterizations is beyond the scope of this observational paper, we have addressed this question by adding a sentence to the SNOWPACK section to describe the arbitrary rate transition, and to refer readers to the work by Lehning et al. (2002b) where this topic is addressed in more detail. We note that we also have text and references elsewhere in the manuscript addressing the relationship between temperature gradients and vapor flux.

References:
Akitaya, E.: Studies on Depth Hoar, Contributions from the Institute of Low Temperature Science, 26, 1–67, 1974.
Colbeck, S. C.: A review of the metamorphism and classification of seasonal snow cover crystals, in: Avalanche Formation, Movement and Effects, edited by: Bruno Salm and Hansueli Gubler, 3–34, 1987.
Lehning, M., Bartelt, P., Brown, B., Fierz, C., and Satyawali, P.: A physical SNOWPACK model for the Swiss avalanche warning Part II. Snow microstructure, Cold Reg. Sci. Technol., 35, 147–167, 2002.

Regarding the atmospheric forcing of the model: is 2-m air temperature or surface temperature used from ERA5? How well are strong near-surface air temperature gradients resolved within the atmospheric boundary layer? How reliable is ERA5 surface temperature?

> These are important calibration and validation questions facing the entire ice sheet community due to the lack of in situ observational data across the Greenland ice sheet which would be required to investigate these issues. We note that numerous studies employing firn modeling use ERA5 as forcing (e.g., Noël & Lipscomb, 2022). Importantly, the validity of our findings does not solely rest on the analysis utilizing ERA5 forcings, which is our motivation for also including analyses of our direct in situ measurements.
References:
B. Noël and W. H. Lipscomb, "Peak refreezing in the Greenland firn layer under future warming scenarios," *Nature Communications*, 2022. doi: 10.1038/s41467-022-34524-x

In the synthetic wetting front simulation, two superimposed sinusoidal curves represent annual and daily cycles in surface temperature. But nowhere does this temperature exceed the melting point. How, then, is surface melt produced?

> Surface melt is produced by the energy inputs stemming from air temperature, incoming shortwave, and incoming longwave radiation. At high elevations in Greenland's percolation zone, it is very common for surface melt to form when temperatures are not above melting point (but close to it), largely due to the heat provided by shortwave radiation. Numerous other studies have confirmed this. Within the model, we refrained from maximum temperature values ≥0°C as they produced anomalously high melt volumes and melt percolating to depths unlike those we have observed in the percolation zone. Thus, we have capped temperature just shy of

the melting point, which is compatible with observations at the Crawford Point meteorological station.

Have you checked the surface melt volume in the model with observations? Does the surface energy budget in SNOWPACK make sense? Would it be possible to plot a time series of surface melt along with figure 7?

> As described in the manuscript, our simulations were designed as synthetic representations of a seasonal wet layer, informed by extensive measurements of wet layer depths and durations. We adopted this approach because no existing model can reproduce the observed data with reliable precision. This limitation arises from inadequate parameterizations of energy fluxes, including mass loss to sublimation, and more critically, the inability to represent meltwater infiltration through preferential flow paths.

> While the amount of melt generated by SNOWPACK falls within a reasonable range, it is not appropriate to use it as a direct comparison series in this context. Our primary focus is on the characteristics of the resulting wet layer, which are realistically captured, as demonstrated by the agreement between Figure 7 (observational data) and Figure 8 (model output). A comprehensive model evaluation lies beyond the scope of this study, which is why we emphasize synthetic model runs. To provide additional context, we have added a reference to Moon et al. (2025), which offers a detailed comparison of models (including SNOWPACK) and in situ observations.
References:
Moon, T., Harper, J., Colliander, A., Hossan, A., and Humphrey, N.: L-band radiometric measurement of liquid water in Greenland's firn: Comparative analysis with *in situ* measurements and modeling, Ann. Glacio., 66, e15, 1-9, https://doi.org/10.1017/aog.2025.10012, 2025.

**(3)** Effectiveness of figures

I find figure 5 quite uninteresting and irrelevant. These temperature profiles are monthly averaged, likely eliminating most TGs exceeding the CTG. In fact, the highest TGs are near the surface, but this is no support for the hypothesis of meltwater-induced TGs. Indeed, only in May is there a temperature gradient exceeding the CTG, but that is before the melt season and not in the location where meltwater-related faceted grain growth occurs. So I suggest replacing this figure by a figure with a few examples of instantaneous observed temperature profiles that illustrate TGs around an observed wetting front, a piping event, both positive and negative. It would be far more elucidating what kind of TGs are typically encountered in a melting firn pack than the current figure 5.

> A distinctive feature of Greenland's percolation zone is the extreme seasonal temperature gradients within the firn column: the summer surface reaches the melting point while the underlying firn retains substantial cold content from the previous winter. Building on prior work discussed in our introduction (e.g., Alley et al., 1990; Steffen et al., 1999), we initially

hypothesized that these seasonal swings could generate strong temperature gradients (CTGs) and promote faceting. However, this figure demonstrates that month-to-month changes in the firn temperature profile are not large enough to drive kinetic grain growth. Importantly, the seasonal reversal of temperature gradients remains a key aspect of our findings. For these reasons, we consider this figure an essential piece of evidence supporting the conclusions of our study.
References:
Alley, R. B., Saltzman, E. S., Cuffey, K. M., & Fitzpatrick, J. J.: Summertime formation of Depth Hoar in central Greenland, Geophys. Res. Lett., 17, 2393–2396, 1990.
Steffen, K., Abdalati, W., & Sherjal, I.: Faceted crystal formation in the northeast Greenland low-accumulation region, J. Glaciol., 45, 63–68, 1999.

As mentioned, are the statistics in figure 6 independent of sample size and record length?
> This comment is addressed in detail in a prior response.

Figure 7 is a great figure! I suggest to explicitly mark the wetting front and piping episodes in the figure, just with a stylized line and word labels in the figure.
> Thank you! The relative range of wetting front position through time is denoted by a grey line. The upper boundary of the wet layer (when not = ice sheet surface position) is also denoted by a grey line. The piping episode is highlighted by a black dashed box in Fig. 7c. Upon your suggestion, additional word labels have been added to the figure for clarity.

Figure 8 nicely shows how the wetting front is wiping out faceted grains. This process, which is casually mentioned in line 227, is crucial in understanding that the actual occurrence of faceted crystals depends on TG around infiltrating meltwater, but also on the presence of meltwater itself. Perhaps interesting to expand this figure with a infographic, or like suggested for figure 7, to annotate panel b with a line of text that explains what is happening.
> While the main text discusses this, we agree with this comment and have made changes to bring the important point to the reader's attention in the figure by expanding the figure caption.

I'm not sure if the division into a main text and a supplement is really necessary for this paper. The paper is not really long, and the normal methods section seems a fine place to put all the information of the supplement into.
> We have moved the text from the previous Supplement that describes the model spin-up and the formulation of the two synthetic model scenarios into the main manuscript. However, we believe it is best to retain the tables and figures in the Supplement, as they provide detailed information on model settings, forcings, and site-specific data. These materials occupy considerable space and are not central to the main discussion, but they are essential for ensuring the reproducibility of our study.

Minor Remarks

L. 9: Be specific here. Instead of "two atmospheric drivers", write something like: "Kinetic grain growth is known to occur when very large temperature gradients cause vapor transport within

the firn. In previous work, warm wintertime winds and summertime absorption of sunlight have been identified to cause such large gradients. Here we demonstrate that meltwater infiltration can also cause such gradients, explaining observations of faceted grains in the percolation zone, and at greater depth."

> The authors agree that this will be a useful addition. Beginning line 8, we have made the necessary additions/deductions to the text, resulting in the following revision: "At high-elevation, dry firn locations, kinetic grain growth is known to occur when very large temperature gradients cause vapor transport within the firn. Previous studies identify warm katabatic winds in winter and solar insolation in summer as drivers of kinetic grain growth, though both produce only millimetre-scale layers near surface. Here, we demonstrate that meltwater infiltration in the percolation zone of the Greenland Ice Sheet (GrIS) can also cause temperature gradients sufficient for kinetic grain growth, explaining our observations of centimetre- to decimetre-scale layers of kinetic grain forms (ranging from faceted crystals to depth hoar) persisting to depths of up to 16 m."

L. 25: The best way of citing Amory et al. is "The Firn Symposium Team, 2024". Charles Amory is lucky to have a name starting with A :-), but no specific author order is implied in the author list of that paper.

> Indeed! The in-text citation(s) for this reference, as well as its corresponding entry in the bibliography, have been changed (L. 25 & 323).

L. 39: Here is a good opportunity to describe in one or two lines what kinetic grain growth precisely is, and how it is physically different from rounding.

> The authors agree. A more descriptive statement regarding what kinetic grain growth is has been added to the text (beginning at line 46 with tracked changes turned on; line 44 with them hidden).

L. 51: remove "later"
> "Later" has been removed from the text.

L. 57: In a separate paragraph, here is a good place to put forward more explicitly your central hypothesis, namely that meltwater is also a cause for enhanced temperature gradient driven kinetic grain growth.

>This is primarily a matter of writing style and space. Our introduction is already relatively long at six paragraphs, and we prefer not to frame the entire paper explicitly as a test of a proposed hypothesis. Instead, we begin by presenting our unexpected observations and then explore a plausible explanation. This approach emphasizes that the observations themselves are significant and stand on their own—given their uniqueness, one could even write a shorter paper focused solely on reporting them. Framing the work strictly around a hypothesis would risk diminishing the intrinsic importance of these observations.

L. 71. The last sentence is more for a conclusions section (and is already mentioned in the abstract too). Better remove here.
> Good point – we have followed this advice and removed the sentence.

L. 290. This deserves some more attention. Is kinetic grain growth by wetting front advance more important / more frequent than by preferential flow? Is there any firn core evidence that the former mechanism explains the majority of faceted crystals?

>Unfortunately, firn core observations alone do not allow us to confidently link faceted layers to specific infiltration mechanisms after the fact—hence our reliance on temperature data analysis and synthetic model runs. We agree that this finding is important. To strengthen this point, we have added text to both the section on facet observations and the end of this section. Specifically, we note that we did not observe faceting along ice pipes in the cores, which is consistent with our modeling results (although a lack of evidence is never proof). We also emphasize that the following section addresses the dominant role of wetting fronts, and the conclusion section reiterates this.
* * *
**Reviewer:** Mahdi Jafari

**Comments to the Author**
General Overview

This manuscript presents a study on **kinetic grain growth in firn induced by meltwater infiltration** on the Greenland Ice Sheet. The authors provide new and interesting observations that extend previous understanding of kinetic grain growth, which has typically been confined to thin, near-surface layers of millimeter-scale thickness. Here, the authors report the presence of faceted layers up to 1 m thick and preserved at depths as great as 16 m, which is a novel and noteworthy finding.

The work is timely and relevant, as it sheds light on the interplay between meltwater infiltration, temperature gradients, and firn microstructure evolution in the percolation zone. The proposed mechanism—that meltwater infiltration advects heat into cold firn and establishes strong local temperature gradients sufficient to drive kinetic grain growth—is plausible and well-motivated. The integration of field observations with SNOWPACK modeling provides a valuable perspective, although some aspects of the methodology and model assumptions need further clarification.

Overall, the manuscript addresses an important gap in the literature by linking microstructural evolution of firn to surface processes, meltwater dynamics, and densification models. With additional clarification and refinement in key sections, this study could make a significant contribution to the understanding of snow and firn processes on the Greenland Ice Sheet.

Major/Minor Comments

1.  It would be helpful if the authors could provide a short description of "kinetic grain growth" in the Methods section. This would give readers a clearer understanding of the physical process that governs kinetic grain growth.

> We agree this is a needed addition, which was also suggested by reviewer 1. We have added an explanation to the introduction because we believe it is best to provide the reader with this up front in the article.

2. **Lines 30–31:** Can you clarify if you are referring to the wind-compaction process observed in Arctic snow covers, which leads to a hard slab forming on top of the depth hoar layer? What is the timescale of forming a sub-millimeter hard slab compared to faceted layers that result from kinetic grain growth?

   This paragraph states that diffusive vapor transport drives the rounding process, leading to hard slab burial several meters deep with rounded grains of ~2 mm radius. Could the authors also check whether other types of water vapor transport, such as convection, may occur in this context?

3. **Lines 42–44:** Can you please clarify the significance of convection-driven water vapor transport in the Greenland Ice Sheet? Jafari et al. (2020, 2022, 2023) have shown that if convection persists over long timescales, it can substantially decrease density at the bottom of the snowpack and increase density at the top. **Could the authors be more explicit about which types of water vapor transport drive the kinetic grain growth rate** and discuss the relevance of convective vapor transport in this context?
   References:
      Jafari, M., Lehning, M., and Sharma, V. (2022), Convection of water vapor in snowpacks. J. Fluid Mech.
      Jafari, M., Gouttevin, I., Couttet, M., Wever, N., Michel, A., Sharma, V., and Lehning, M. (2020), The impact of diffusive water vapor transport on snow profiles in deep and shallow snow covers and on sea ice. Frontiers in Earth Science.
      Jafari, M., Lehning, M. (2023), Convection of snow: when and why does it happen? Frontiers in Earth Science, 11, 1167760.

> We address these three comments together because they relate to the same two-paragraph section of the introduction. The purpose of these paragraphs is to provide a general overview of firn evolution along the flanks of the Greenland Ice Sheet, as described in the literature. Our intent is to establish that the typical trajectory of firn is grain rounding and densification as it transforms into glacier ice over decades. The formation of faceted grains represents an unusual, transient deviation from this overall process of rounding, compaction, and transformation. Our intention with these two paragraphs is to offer a concise, well-referenced summary of this context.

> Regarding slabs: While firn columns may contain layers of varying density, the term "slab" is not commonly applied to polar ice sheets as it is in seasonal snowpacks. Unlike seasonal snow, there is no ground heat flux to produce thick basal facets (given the absence of ground and the effectively infinite thickness of the firn), and one could argue that the entire firn column behaves as a hard slab. This setting is fundamentally different from Arctic seasonal snowpacks, but we do not have the space to provide a detailed comparison with all other environments.

Regarding diffusion and convection: Our paper does not aim to address the detailed physics of grain metamorphism. Instead, we adopt the well-established premise that kinetic growth is linked to the thermal field (see references). We refer readers to the relevant literature for a comprehensive discussion, as we do not have the space—nor do we offer new insights—on these processes. To acknowledge that convection may influence vapor transport, we have added text and references to the papers listed above.

4. **Lines 80–83:** For the lower elevation, the temperature measurements show that the summer heatwave penetrates more deeply. The manuscript explains this as (1) due to increased latent heat — is this because of warmer air at lower elevations? — and (2) due to higher thermal conductivity from greater firn density. Could the authors clarify which of these is the main driver? Also, what is the reason for the larger density at lower elevation — could this be related to stronger katabatic winds favoring wind compaction?

> First, to answer the questions: Lower elevations have warmer temperatures and fewer clouds and thus greater melt rates. The result is higher heat input to the firn column from both conductive and latent heat transfer associated with refreezing of meltwater. The firn has higher bulk density due to a greater fraction of ice from refreezing meltwater and high densification rate of the firn fraction under warm conditions. While the increased density of firn at lower elevations does influence conductive heat transfer, the latent heat transfer dominates the heat balance.

The above issues are discussed in detail, with analyses and evidence, in two papers: 1) Saito et al. (2024) is focused on heat transfer, and 2) Harper et al. (2012) is focused on density changes. While we do not have space to fully present the above issues and provide evidence, we have added text with references to the manuscript to clarify questions readers may have.
References:
Harper, J., Humphrey, N., Pfeffer, W. T., Brown, J., and Fettweis, X.: Greenland ice-sheet contribution to sea-level rise buffered by meltwater storage in firn, Nature, 491, 240–243, https://doi.org/10.1038/nature11566, 2012.
Saito, J., Harper, J., and Humphrey, N.: Uptake and Transfer of Heat Within the Firn Layer of Greenland Ice Sheet's Percolation Zone, J. Geophys. Res. Earth Surf., 129, https://doi.org/10.1029/2024JF007667, 2024.

5. **Section 2.4, SNOWPACK simulations:** Can you please clarify whether you used a SNOWPACK version that includes the vapor transport scheme? If yes, could you provide a comparison between the vapor flux calculated from the measured temperature gradients and the flux simulated by SNOWPACK?
> Yes, we did run SNOWPACK with "mass transport by vapor flux" turned on for all simulations (see section 2.4). Vapor fluxes output by SNOWPACK and those we calculated from our measured temperature data were on the same order of magnitude (1e-7). They were also on the same order of magnitude as those reported by Sturm & Benson (1997) in a cold subarctic snowpack. Fluxes modelled by SNOWPACK are not featured in figures, but statements

regarding their relative magnitude in comparison to those we calculated from measurements have been added to section 3.3.

References:

Sturm, M. and Benson, C. S.: Vapor transport, grain growth and depth-hoar development in the subarctic snow, J. Glaciol., 43, 42–58, https://doi.org/10.3189/s0022143000002793, 1997.

6. **Line 158:** Please correct the citation "with mass transport by vapor flow (Lehning et al., 2002a)". The more recent reference for SNOWPACK with water vapor transport is: Jafari, M., Gouttevin, I., Couttet, M., Wever, N., Michel, A., Sharma, V., and Lehning, M. (2020), The impact of diffusive water vapor transport on snow profiles in deep and shallow snow covers and on sea ice. Frontiers in Earth Science.

> The citation has been changed in-text and added to the bibliography

7. **Lines 162–164:** Regarding the idealized simulations, it is not clear how using meteorological forcing with an annual wavelength is justified. Please elaborate on this assumption, especially since it is explicitly stated that a mean annual value is set for the firn temperature.

> The spin-up for the synthetic simulations was the only part of modelling methodology that employed annual-wavelength meteorological forcings. We chose these forcings as to generate a simple, generic firn column to perturb, with thermal conditions like that of the study area. The MAAT sentence has been re-written for clarity to address your comment -- what we meant by this is that the MAAT on the SINUSOIDAL temperature curve is equal to the MAAT at ~2000m; it was not a constant throughout the spin-up.

8. **Line 167, Table S2:** Is LWR a net value (i.e., LWR = ILWR − OLWR, where ILWR = incoming longwave radiation and OLWR = outgoing longwave radiation)? It is not clear what is meant by "heat fluxes during the spinup to retain a vertical deep-firn temperature profile at −16.5°C." Could the authors clarify this statement?

> The text in the supplement is now incorporated into the main text, alongside relevant changes, per request of Reviewer 1. Both longwave and shortwave radiation values in SNOWPACK are incoming. We agree that our text below Table S2 was unclear. Furthermore, this comment caused us to discover that we had included an older description of this methodological step in the SI after we changed our approach. We have updated the text and SI to be both more clear and correct. To summarize here, the constant LWR value we used in the spin-up (300 W m$^{-2}$) was manually prescribed, as it allowed for the temperature profile at depth, which is isolated from seasonal temperature changes, to be nearly vertical and stable at -16.5°C. The reason for this: observations of deep-firn temperature profiles in our study area show nearly vertical, constant temperatures at depth, roughly equal to the MAAT at a site (Saito et al., 2024). If the spin-up did not produce a nearly-vertical temperature profile at depth, this meant to us that 1) the temperature profile had not reached a quasi steady-state condition and 2) that it did not reflect typical conditions at the site. Thus, LWR was set to this value to "balance" heat fluxes in and out of the modelled firn column and produce a vertical temperature profile by the end of the 50-year spin-up.

References:

Saito, J., Harper, J., and Humphrey, N.: Uptake and Transfer of Heat Within the Firn Layer of Greenland Ice Sheet's Percolation Zone, J Geophys. Res. Earth. Surf., 129, https://doi.org/10.1029/2024JF007667, 2024.

9. **Lines 169–171:** Can you please confirm that the data were averaged from 1995 to 2020 (25 years) at each time step (hourly resolution)? Does this averaging then provide the time series for air temperature, shortwave radiation, and longwave radiation used as forcing data?

> We did indeed average the ERA5 temperature and SWR values at every hour on every day of the year over the 25-year period, a common approach to initialization for firn column models. So, each average hourly value was calculated from ~25 datapoints. From these time series of averages, we created smoothed, generalized temperature and SWR forcings as seen in Figure S1 (a, b, orange curves) and (c, grey curve). These curves are what the synthetic models were forced with. LWR forcings were calculated from temperature (Figure S1 c (grey curve) and (d)) using the Stefan-Boltzmann Law.

10. **Lines 254–256:** This appears to contradict what I observe in Figure 8, where the sphericity increases after reaching its maximum reduction. Could the authors clarify and be more specific about the sphericity reduction shown in Figure 8?

> This is a good point: the statements regarding sphericity changes seen in Figure 8 (b) were misleading. The **strongest** sphericity reductions occurred prior to and during wetting front descent (and were later overprinted), whereas those sphericity reductions that did occur beneath the maximum wetting front extent resulted in less-faceted grains that did not persist in the firn column. We have adjusted wording to more clearly depict these interpretations.

11. **Line 281:** It seems that the authors are referring to a different plot, not Figure 7, since Figure 7 is about wet layer onset as the faceting mechanism. Could the authors clarify this reference?

> Figure 7 showcases both wet layer onset (lefthand column) and piping (righthand column) as faceting mechanisms, therefore the reference to Figure 7 is correct.

12. **Caption of Figure 7:** I think both columns should belong to the same simulation setup. Please specify which faceting mechanism is shown — it should be the wet layer onset, as this plot corresponds to Section 3.3. What causes the transient change seen in the black box in Figure 7(c)? Is it a numerical artifact or a physical process? Could the authors add another row showing the time series of grain types, which would be very informative for readers? SNOWPACK simulations include grain type as an output that can be used for post-processing.

> The results shown in all subplots in Figure 7 are measured/calculated time series (field measurements) at site CP-1998m in 2023. The black box in 7(c) is an inset that shows the location of the data in the righthand column. The data within the black box shows the supposed thermal signature of a preferential flow event. Unfortunately, we do not have a corresponding time series of measured grain type or sphericity at this site; we drill one firn core in the late spring and are unable to drill cores all summer long.

13. **Line 287:** Please indicate that subplots (E) and (F) of Figure 8 correspond to the "Preferential flow/piping" scenario. Change the reference from "(Fig. 8)" to "(Fig. 8 E and F)".

> Good point - the reference has been changed to "(Fig. 8 E and F)".

14. **Lines 316–319:** This is very important regarding densification models. Faceted grains have a very different densification pattern compared to dry firn conditions. Do the SNOWPACK simulations include densification specifically elaborated for faceted grains? If not, please mention this limitation for snow models such as SNOWPACK.

> We agree this is very important – a primary motivation for the paper, and why we treat this with a paragraph in the discussion and highlight this again in the conclusions. Firn densification is very important to various aspects of ice sheets, from dH/dt measurements to ice core gas exchange. While SNOWPACK does consider the differing mechanical behaviors of faceted snow grains, what is more important in this discussion are the models researchers use to simulate long term densification of polar firn (e.g., Herron and Langway).

References:

Herron, M. M. and Langway, C. C.: Firn Densification: An Empirical Model, J. Glaciology, 25, 373–385, https://doi.org/10.3189/s0022143000015239, 1980.